# Anthropometric and Physiological Profiles of Hungarian Youth Male Soccer Players of Varying Ages and Playing Positions: A Multidimensional Assessment with a Critical Approach

**DOI:** 10.3390/ijerph191711041

**Published:** 2022-09-03

**Authors:** Imre Soós, Krzysztof Borysławski, Michał Boraczyński, Ferenc Ihasz, Robert Podstawski

**Affiliations:** 1Faculty of Health Sciences, Doctoral School of Health Sciences, University of Pécs, H-7622 Pécs, Hungary; 2Institute of Health, Angelus Silesius State University, 58-300 Wałbrzych, Poland; 3Faculty of Health Sciences, Collegium Medicum, University of Warmia and Mazury in Olsztyn, 10-719 Olsztyn, Poland; 4Faculty of Psychology and Pedagogy, Institute of Sports Sciences, Eötvös Lóránd University, 9700 Szombathely, Hungary; 5Department of Tourism, Recreation and Ecology, University of Warmia and Mazury in Olsztyn, 10-957 Olsztyn, Poland

**Keywords:** soccer, men, age categories, anthropometric features, physiological characteristics

## Abstract

Background: This study aimed to create preliminary anthropometric and physiological profiles of Hungarian male soccer players belonging to different age categories (14, 15, 16, and 17–18-years) and assigned to different playing positions (forward, defender, midfielder, goalkeeper). Methods: Anthropometric and physiological profiles were created for four age groups: 14- (*n* = 20), 15- (*n* = 16), 16- (*n* = 22) and 17–18-year-olds (*n* = 23) representing the Hungarian soccer academy. Additionally, the variables were analyzed across the four player positions mentioned above. Results: The mean values of body mass, fat mass and BMI were within normal limits, although in some cases the anthropometric and body composition values were too high, particularly among the 17–18-year-olds. The mean values of HR_rest_ were lowest among the 15-year-olds. The highest mean and maximal values of rVO_2max_ and rVO_2_/AT [57.6 ± 8.12 (43.8–68.3) and 51.2 ± 7.24 (38.9–60.8) mL/kg/min, respectively] were noted in 14-year-olds. Goalkeepers performed significantly better than the remaining soccer players in terms of the most anthropometric and physiological characteristics, except for the Yo-Yo test (*p* < 0.001). Conclusions: The values of anthropometric parameters increased with age. As expected, the oldest group achieved the best results in the performance tests. Goalkeepers outperformed the players representing other playing positions in the tests when assessing lower limb strength, sprint performance (5- and 10-m distance), and agility tests. From a practical point of view, the presented anthropometric and physiological profiles of players representing different age groups and playing positions can be useful for soccer coaches, strength and conditioning specialists, and athletic trainers of other soccer clubs in terms of the individualization and optimalization of soccer training.

## 1. Introduction

Soccer is a complex sport that consists of many different activities, including short sprints, rapid acceleration and deceleration, tackling, turning, jumping, and kicking [1,2,3]. It is considered a high-intensity intermittent team sport due to its acyclic nature and dynamically shifting exercise intensity [4]. In the course of a match, soccer players engage in a variety of acyclic activities at differing intensities; thus, soccer training should incorporate exercises to develop all aspects of physical performance [5]. Concomitant increases in both the number of games in a season and the game pace have placed intense demands onto the players’ motor skills [6,7]. This has important implications for sports training and increasing training loads should be applied in the process of enhancing motor skills [8].

The physical factors associated with successful soccer performance have been well described [9]. Whilst improved high-intensity running capacity has been shown to distinguish between low- and highly-trained players [10], other skills that require increased anaerobic capacity and neuromuscular power, such as sprints, jumps, duels, and kicking have also been shown to discriminate between different levels of soccer players. For instance, Vaeyens et al. [11] observed better performance on skills requiring increased anaerobic power (sprint, vertical jump and standing broad jump) in elite youth soccer players when compared with sub-elite and non-elite youth soccer players. Little is known about the age-related variation in anaerobic performance in elite youth soccer players. Although the scientific evidence is scarce, some studies have examined match running characteristics among youth players [12]. Their results indicate that U13–U18 players’ physical match characteristics are affected by positional demands, as strikers and wide midfielders demonstrated the highest peak game speeds and frequency of high-intensity actions [13,14]. Additionally, center midfielders have been reported to cover the highest total distance, whereas center backs covered the lowest total distance and performed the lowest number of high-intensity bouts [15].

The performance profiles of players and teams can be influenced by both biological and environmental factors [16]. According to Mikulić [17], precise information on the anthropometric and physiological status of athletes is important for: (a) providing detailed insights into the status of individual athletes and designing effective training programs, and (b) selecting athletes because certain characteristics, such as anthropometric measurements of length and breadth, are almost entirely genetically conditioned and thus show little or no response to training programs. Anthropometric, physiological, and physical capabilities that are closely related to successful soccer performance play a particularly important role in the assessment process.

There is evidence to suggest that specific physiological demands and anthropometrical prerequisites are associated with different playing positions; therefore, young players should be selected on the basis of superior physiological performance and anthropometric advantage [18]. Considerable research has been done into the anthropometric and physiological profiles of adult soccer players [1,19,20], but fewer studies have examined the anthropometric and physiological characteristics of young soccer players representing different age categories and playing positions [21]. Moreover, these characteristics have not been investigated in the context of competitive soccer in Hungary, and there is a general scarcity of literature on the anthropometric and physical performance characteristics of Hungarian soccer players. The concept of creating anthropometric and physiological profiles using multi-faceted tests with a critical approach is very timely because talented soccer players are difficult to identify. This is due to the fact that the development of young players is determined by numerous factors, including their anthropometric, physiological, technical, tactical, and psychological characteristics, as well as environmental and sociological influences [22,23]. From this standpoint, we believe that research into these aspects of Hungarian soccer players could provide valuable insights for soccer coaches, strength and conditioning specialists, and athletic trainers and, as a result, improve the effectiveness of youth training.

Therefore, this study had two objectives: (a) to develop the anthropometric and physiological profiles of Hungarian male soccer players belonging to different age categories (14-, 15-, 16- and 17–18-years old) and playing different positions, and (b) to identify and explain potential differences between the analyzed groups of athletes who have not yet achieved the highest level of soccer performance.

Due to the fact that match running performance is age-dependent, the existing knowledge gap should be filled to assist practitioners in determining the age(s) at which additional training is necessary to develop the running ability for the most physically demanding positions. The following research question was formulated to address these issues: are there any practically relevant differences in the anthropometric and physiological characteristics between playing positions in soccer players across youth categories?

## 2. Materials and Methods

### 2.1. Participants

This study was conducted at the Rába ETO FC soccer club (Győr, Hungary) during an off-season period (January–early February 2022). A total of eighty-one competitive, high-level male soccer players (aged 14–18 years) were recruited to take part in this cross-sectional study. All players belonged to the Fehér Miklós Elite Football Academy and represented top-ranked Hungarian male soccer teams (first-division championship, MLSZ league). Competition age categories were used as subgroup allocation criteria, and four subgroups were created: 14-year-olds (*n* = 20), 15-year-olds (*n* = 16), 16-year-olds (*n* = 22), and 17–18-year-olds (*n* = 23).

Across all groups, a given playing position was represented by the following number of players: forwards (*n* = 23), central defenders (*n* = 15), external defenders (*n* = 12), central midfielders (*n* = 12), external midfielders (*n* = 10), and goalkeepers (*n* = 7). The analyzed subgroups (14-, 15-, 16-, and 17–18-year-olds) had 4.2 ± 0.6 years, 5.4 ± 0.9 years, 6.1 ± 0.8 years, and 7.8 ± 1.5 years of experience in structured soccer-specific training, respectively. The two youngest subgroups (14- and 15-year-olds) trained for 90 min three times per week (Monday, Wednesday, Friday) and played an 80-min control match during the weekend (alternating Saturday or Sunday). The two older subgroups trained for 90–120 min four times per week (Monday, Tuesday, Thursday, Friday) and played an 80-min control match during the weekend (Sunday). All the players, irrespective of age groups, participated on average in ~12 h of combined soccer-specific training, competitive control match, strength, coordination, and conditioning training sessions. Players were excluded from the study in the following cases: (I) acute injury or a condition that limited physical function; (II) lower limb injury in the past six months.

### 2.2. Ethical Statement

The participants and parents or legal guardians gave their written informed consent to participate in the study and to publish the players’ images, in accordance with the ethical guidelines of the Declaration of Helsinki and subsequent updates. This study was approved by the Ethics Committee of the National Public Health Center (NPHC) (17990-7/2022/ECIG).

### 2.3. Procedures, Data Collection and Equipment

The anthropometric, physiological and performance characteristics of youth soccer players from the priority academies are continuously measured according to the instructions of the Ministry of Human Resources. The tests are always carried out by the same trained professionals. The data are entered into a central database, which is a cloud-based platform for athletes (TalentX). Thus, these data were analyzed in this study. The participants’ demographic and anthropometric data were collected in the first stage of the study. Biological age was estimated based on morphological age (MA), according to the method described by Mészáros and Mohácsi [24]. In brief, the mean of the chronological age (CA) and three age equivalents of body height (BH), body mass (BM) and the plastic index (PLX), each determined to the nearest 0.25 years, were calculated according to the following formula: MA = 0.25 − (BH age + BM age + PLX age + CA). A standard 8-min warm-up was then performed (running, joint mobility exercises, and neuromuscular activation exercises). The results of the following assessments were then recorded: isometric strength of hip abductor (ABD) and adductor (AD) muscles, and the Y Balance Test (YBT). All tests were verbally explained, and the participants performed a practice round to become familiar with the test procedure (the results were not recorded). Data were collected and uploaded to a digital platform by two members of the research team.

#### 2.3.1. Anthropometry

All anthropometric characteristics were measured by a trained ISAK-accredited anthropometrist (Level 1) in accordance with the standardized procedures of the International Society for the Advancement of Kinanthropometry (ISAK). Body height (BH) was measured to the nearest 0.1 cm using a patient weighing scale with a height rod (Seca 217, Hamburg, Germany). Body mass (BM), after the removal of shoes and heavy clothing, was measured to the nearest 0.1 kg. Body composition variables (percentages of fat mass and muscle mass) were measured in the standing position via the InBody 720 Tetrapolar 8-Point Tactile Electrode System (Biospace Co., Ltd., Seoul, Korea). Body composition measurements were performed in accordance with the relevant measurement guidelines [25].

Three body dimensions (shoulder width, lower arm girth, and hand circumference) were measured using specialist anthropometric equipment (Martin Anthropometer, GBM, SiberHegner, Zurych, Switzerland, 2003) and a metal measuring tape (Holtain, Crymych, UK). The results were used to calculate the plastic index (PLX) as the arithmetic sum of three body dimensions that are characteristic of bone-muscle development. The following formula was used: PLX (cm) = shoulder width (cm) + lower arm girth + hand circumference (cm). The numeric values of these indices can be used to create a right-angle coordinative system, where the vertical axis is scaled by the metric index, and the horizontal axis refers to the plastic index. The metromorphic-normoplastic body build is located at the center of the coordinated system. The upper-left quarter contains leptomorphic-hypoplastic individuals, and the right-upper quarter refers to the leptomorphic-hyperplastic body build. The lower-left area is characteristic of pycnomorphic-hypoplastic individuals, and the lower-right quarter contains pycnomorphic-hyperplastic physique variants. In children, the vertical axis should be positioned at the level of the respective PLX averages.

#### 2.3.2. NordBord Hamstring Testing System Device (VALD Performance Pty Ltd., Brisbane, Australia)

The muscles in the posterior compartment of the thigh are collectively known as the hamstrings. They consist of the biceps femoris, semitendinosus and semimembranosus. The participants knelt over a padded board, with the ankles secured immediately superior to the lateral malleolus by individual ankle braces that were attached to custom-made uniaxial load cells (Delphi Force Measurement, Gold Coast, QLD, Australia) with wireless data acquisition capabilities. Following a warm-up set, the players performed one set of three maximal repetitions of the bilateral Nordic hamstring exercises. They were instructed to gradually lean forward at the slowest possible speed, while maximally resisting the movement with both legs, holding the trunk and hips in a neutral position, and crossing their arms across the chest throughout the exercise [26]. The participants were loudly exhorted to provide maximal effort during each repetition. A trial was considered acceptable when the force output reached a distinct peak (indicative of maximal eccentric knee flexor strength), followed by a rapid decline in force, which occurred when the athlete was no longer able to resist the effects of gravity acting on the segment above the knee joint. Peak eccentric knee flexor strength was determined based on the maximum force of three trials (N), and it was converted to joint torque (N/m) based on the right shank length (lateral tibial condyle to lateral malleolus).

#### 2.3.3. ForceFrame Strength Testing System Device (VALD Performance, Albion, Australia)

Hip strength procedure: The isometric strength of the hip ABD and AD muscles was measured using the ForceFrame Strength Testing System device (VALD Performance Pty Ltd., Brisbane, Australia), based on a previously described protocol [27]. For the implementation, the participants were asked to lie in a supine position under the ForceFrame system, with the femoral condyle of their knees on the padded load cell (100 Hz) at the 60° angle (short lever, hips also bent at 60°). The height of the bar was adjusted to each player to ensure that the specific angle was maintained. The players were first given a demonstration by the investigators and then they performed 1–2 warm-up practice repetitions. The participants were first asked to perform an isometric contraction of the hip AD for 5 s and, following a 5-s rest period, a 5-s isometric contraction of the ABD. After a 45-s rest period, the same procedure was repeated, the results were recorded, or a second attempt was made. Hip ABD and AD strength was determined based on the maximum peak force of three trials (N). These values were subsequently converted to joint torques (N/m) based on the right leg length (anterior superior iliac spine to lateral malleolus). The results of the isometric contraction tests were used to calculate two additional parameters: the ratio of hip AD/ABD strength, calculated for each leg using the following formula: hip AD strength/hip ABD strength of the homolateral leg; and relative bilateral strength asymmetry (the strength imbalance between the limbs, both AD and ABD), with the following formula: [(dominant leg muscle strength − non-dominant leg muscle strength)/dominant leg muscle strength] × 100 [28].

#### 2.3.4. Sprint Performance Linear Sprinting (5, 10, and 20 m)

All tests were conducted indoors on a PVC running surface. Four pairs of wireless single-beam timing gates (TAG Heuer, La-Chaux-de-Fonds, Switzerland) were placed 5 m apart. The gates were adjusted to a height of 1 m that approximately matched the participants’ hip height. A high-speed camera (Weinberger Deutschland GmbH, Erlangen, Germany; 100 frames per second) was positioned behind the initial timing gate aligned with the timing light beam. The subject and the timing light beam were within the camera’s field of view. A reflective marker was placed on the participants’ left hip representing the height of the center of mass. The subjects performed a 10-min general warm-up, including light jogging, short accelerations, and dynamic stretching exercises before a familiarization trial. The order of the starting distances was randomized to eliminate the potential effects of fatigue. The rest period between sprints with the same starting distance was 90 s, and the rest period between different starting distances was 3 min. The participants were asked to sprint past a cone placed 1.5 m behind the second timing gate to avoid a finishing dip or deceleration. The subjects were instructed to perform a split start and were allowed to choose the leading foot; however, the same starting position was required in all trials. A signal was not given. Rocking or leaning back before sprinting was not allowed. Timing began when the initial timing gate was triggered. The video sequences were processed using Vicon Motus version 9.2 (Vicon Peak, Los Angeles, CA, USA). The accuracy of the initial timing gate (time [seconds]) was defined as the number of frames between the frame when the light beam first appeared in the video image and the frame when the reflective marker passed the timing gate line divided by 100, and could be interpreted as a timing gate error. The adjusted 5-m time was calculated by subtracting the timing gate error from the 5-m time gained by the timing lights.

#### 2.3.5. 505 Change-of-Direction Test

The methodology for the 505 change-of-direction test was based on the previously described protocols [29]. The subjects performed a standing start with the same body position as in the 30-m sprint, with the front foot 30 cm behind the start line. The subjects sprinted through the timing gate to the turning line marked on the laboratory floor. The subjects were instructed to place the left or the right foot, depending on the trial, on or behind the turning line before sprinting back through the gate. Three trials were recorded for turns off the left and right side, in random order, with a 3-min recovery period between trials. A researcher was positioned at the turning line, and if the subject changed direction before hitting the turning point, or turned off the incorrect foot, the trial was disregarded and repeated after the recovery period. The time for each distance was recorded to the nearest 0.001 s. The mean value from three 505 trials for each leg was used in the analysis. The side with the fastest time was defined as the preferred side, and the other side was regarded as the non-preferred side.

#### 2.3.6. The Illinois Change of Direction (COD) Test (IAGT)

The COD IAGT is set up with 4 cones forming the agility area. On command, the athlete sprints 9.2 m, turns, and returns to the starting line. After returning to the starting line, he swerves in and out of the 4 markers, completing two 9.20-m sprints to finish the agility course [30]. Performances were recorded using an electronic timing system (TC-System, Brower Timing Systems, Draper, UT, USA). Infrared timing gates were positioned at the start and finish lines at a height of approximately 1 m. The top speed from three trials was used in the statistical analyses.

#### 2.3.7. The Yo-Yo Intermittent Recovery Test Level 1

The Yo-Yo IR1 was conducted according to the method described by Krustrup et al. [31]. Twenty-meter shuttle runs of increasing velocity were performed until fatigue, with a 10-s active recovery period (2 × 5 m of jogging) between each run. The test ended when the objective criteria (two failures to reach the front line in time) or subjective criteria (the participant was unable to continue at the required velocity) were met. The total distance covered during the test was used in the statistical analysis.

#### 2.3.8. Cardiorespiratory Testing

The cardiorespiratory test was conducted at the Academy’s Laboratory of Exercise Physiology, using the Piston instrument (EN ISO 13485:2016, Budapest, Hungary). Ergospirometric tests were performed before the beginning of the competition season, following a progressive intensity protocol until voluntary exhaustion (failure) on a motorized treadmill (Pulsar 4.0, h/p/Cosmos Sports & Medical GmbH, Nußdorf, Germany). All players participated in the test after two light soccer adaptation training sessions to minimize injuries. Resting heart rate (HR_rest_) was measured in a laboratory setting by averaging the data for the last 5 min of 20-min sitting resting records. Before starting the workload, the players performed an individual warm-up consisting of 5 min of self-paced cycling and 3 min of dynamic stretching. The test protocol began with an initial speed of 5 km/h (walk) for one minute and continued at 8 km/h. Thereafter, the treadmill speed was increased by 2 km/h every two minutes with a continuous inclination of 2°. The players were instructed to run to exhaustion and were given strong verbal encouragement to perform at their best during the test. Respiratory gas exchange was monitored continuously with a portable breath-to-breath gas analyzer (PRE-201/cc and PRE-201/pm). The analyzer was calibrated according to the manufacturer’s instructions before each trial run. The following cardiorespiratory variables were monitored: heart rate (HR bpm), oxygen uptake (VO_2_ measured in L/min), carbon dioxide production (VCO_2_ L/min), respiratory exchange ratio (RER, arbitrary units; AU) expressed as the ratio of two metabolites (VCO_2_/VO_2_), minute ventilation (VE L/min), and relative oxygen uptake (rVO_2_) at the anaerobic threshold (AT) (rVO_2_/AT mL/kg/min). The anaerobic threshold pulse (ATP) was determined after the completion of the exercise test; ATP was determined for each subject using the V-slope method developed by Beaver et al. [32]. This method involves an analysis of the VCO_2_/VO_2_ response on the assumption that the threshold value corresponds to the breakpoint of the VCO_2_/VO_2_ relationship and the corresponding HR. The values of VE, VO_2_, VCO_2_, and RER were averaged over 10-s periods, and the highest 30-s value (i.e., three consecutive 10-s periods) was used in the analysis. Heart rate (HR) was continuously monitored (at 5-s intervals) before and during the test using a chest transmitter and receiver (Garmin HRM3-SS, Garmin Ltd., Olathe, KS, USA). The VO_2max_ value was accepted if at least three criteria were met: (1) HR in the last minute exceeded 90% of the subject’s age-predicted HR_max_, which was calculated previously with the use of the equation proposed by Tanaka et al. [33]; (2) the VO_2_ plateau reached ≤150 mL/min with an increase in power output; (3) RER reached or exceeded 1.1 AU, and the subjects were unable to continue running despite verbal encouragement [32].

### 2.4. Statistical Analysis

The examined variables were analyzed at four one-year age intervals, and only the last interval covered a two-year period (17–19). The players were also analyzed in four position categories [forward (1), defender (2), midfielder (3) and goalkeeper (4)], but Z-scores were used due to the low number of participants in age and position groups. This approach supported the elimination of age-related differences and a comparison of groups regardless of age. Theoretically, the mean Z-score should be zero (and SD = 1). If the mean Z-score of a given variable was greater than zero in a given category of players, it was assumed that the mean value of this variable in these players exceeded the average. By the same token, if the mean Z-score was less than zero, it was assumed that the mean value of this variable in this category of players was below the average. The homogeneity of variances was checked using Levene’s test. The arithmetic means of the examined variables in the analyzed age and position categories were compared by one-way analysis of variance (ANOVA). Tukey’s HSD test was used in the post-hoc analysis. Post-hoc power was computed for the F tests and one-way ANOVA, given 0.05 of alpha, and 0.40 effect size f, using G*Power version 3.1.9.7 [34]. The achieved power was 0.85. Since this analysis is based on the average group size, the omega-squared (ω^2^) effect size was calculated to test for practical significance. The statistical significance was set at *p* < 0.05. All data were processed with the use of SPSS 25.0 software (SPSS Inc., Chicago, IL, USA).

## 3. Results

### 3.1. Analysis I—Differences in Anthropometric Parameters between Age Groups

In all age groups, the players’ chronological age was highly similar to their biological age. Differences in anthropometric characteristics across the age groups were consistent with the participants’ chronological and biological age. In the vast majority of cases, the highest values of somatic characteristics, including body height, body mass, BMI, and fat mass, were noted in the oldest group. In general, the anthropometric parameters were significantly higher (*p* < 0.001) in the group of 17–18-year-olds than in the 14–15-year-olds, whereas no significant differences (*p* > 0.05) were found between the 16-year-olds and 17–18-year-olds. Percentage muscle mass (%MM) was the only exception, and this parameter was highest in 15- and 16-year-olds, but the differences across the age groups were not significant (*p* > 0.05). In all age groups, maximal body mass was relatively high for soccer players (range: 80.25–91.80 kg). An analysis based on Sheldon’s classification of body types did not reveal significant differences (*p* > 0.05) in the percentage of endomorphic, mesomorphic and ectomorphic somatotypes between groups. However, ectomorphs were predominant in each age group.

### 3.2. Analysis II—Differences in Physiological Parameters between Age Groups

In the vast majority of motor tests (Table 1), 17–18-year-olds scored the best results, which significantly exceeded (*p* < 0.001) those noted in the 14–15-year-olds. Moreover, the oldest players also scored significantly better results (*p* < 0.001) than the 16-year-olds in the 505 COD test for the dominant leg, and the Illinois test with and without a ball [s]. In turn, the results of NB-IBM [%], FF ADD limb [%], and 5-m and 10-m speed [s] tests did not differ significantly (*p* > 0.05) across the age groups. HR_rest_ values were highest in 16-year-olds and 17–18-year-olds (76.3 and 77.5 bpm, respectively), and they were significantly higher than in the group of 15-year-old players. In turn, the mean HR_rest_ values in 15-year-olds (63.5 bpm) were significantly lower (*p* < 0.001) than in the remaining age groups. HR_max_ values in ergospirometric tests, as well as ATP values, did not differ significantly between the 16-year-olds and 17–18-year-olds (range: 194.2–196.7 and 180.6–183.2 bpm, respectively). The mean values of rVO_2max_ and rVO_2_/AT were highest in 14-year-olds (57.6 and 51.2 mL/kg/min, respectively), and they significantly exceeded the values noted in the 16-year-olds (52.6 and 46.8 mL/kg/min, respectively).

### 3.3. Analysis III—Anthropometric Characteristics—Differences between Playing Positions

An analysis of the anthropometric profiles of soccer players assigned different positions revealed significant differences in biological age. Goalkeepers were biologically oldest (0.9) and significantly older (*p* = 0.015) than midfielders (−0.4). Goalkeepers were significantly taller and heavier (*p* < 0.001) than other players. Goalkeepers also differed significantly in terms of BMI (higher BMI relative to midfielders, *p* = 0.011), endomorphic body type (more prevalent than among defenders and midfielders, *p* = 0.043), and PLX (higher PLX relative to forwards and midfielders, *p* < 0.001). An analysis of anthropometric characteristics, including fat mass, muscle mass, and mesomorphic and ectomorphic body type, did not reveal significant differences across playing positions (Table 2).

### 3.4. Analysis IV—Physiological and Performance Characteristics—Differences between Playing Positions

An analysis of the physiological and performance characteristics revealed that goalkeepers received significantly higher scores than other players, in particular midfielders, in the following tests: NN max dominant and non-dominant leg (*p* = 0.010 and *p* = 0.040, respectively), NB avg dominant leg (also higher than in midfielders, *p* = 0.048), FF max dominant ADD (also higher than in forwards, *p* = 0.018), FF max non-dominant ADD (higher only in comparison with forwards, *p* = 0.038), FF max dominant and non-dominant ABD (also higher than in forwards and defenders, *p* < 0.001 for both), 5 m speed (higher only in comparison with forwards, *p* = 0.040), 10 m speed (also higher than in forwards), 20 m speed (also higher than in forwards, *p* = 0.051), Illinois test with a ball (also higher than in forwards and defenders, *p* < 0.001), Illinois test without a ball (also higher than in forwards, *p* = 0.013, and the Yo-Yo test (also higher than in forwards and defenders, *p* = 0.004). No significant differences were found in the values of HR_max_, ATP, VO_2max_ or rVO_2_/AT (Table 2).

## 4. Discussion

The anthropometric and physiological profiles of young soccer players of different ages and playing positions were assessed based on various performance factors. According to Buekers et al. [30], this approach should be adopted to identify youth soccer players with the highest potential for elite performance. The conducted analyses revealed significant differences in the tested variables across groups.

### 4.1. Anthropometric Profile

When the results were compared across the age groups, the highest values of the investigated anthropometric characteristics were noted in the oldest players (17–18 years), which is generally consistent with ontogenetic patterns. Similar results were reported in a study of 12- to 19-year-old top Spanish soccer players, where the basic anthropometric parameters were determined in the following ranges: body mass—58.0–70.0 kg, body height—166.0–175.1 cm, BMI—20.9–22.5 kg/m^2^, endomorphy—2.4–2.9, mesomorphy—3.7–4.1, ectomorphy—2.6–3.0 [1]. Studies evaluating the relationship between match running performance and anthropometric parameters (body height and mass, skinfolds) in young soccer players demonstrated that excessive body mass and excessive fat mass were bound by weak to moderate correlations with subscapular and abdominal skinfolds, which is undesirable [35,36]. For this reason, the maximum body mass values in 16-, 17- and 18-year-old Hungarian soccer players (which exceeded 91 kg in some cases) gives serious cause for concern. However, the average values of BMI and %BF were within normal limits in all age groups. The above individual results are somewhat surprising because the tested subjects participated in a special training program modeled on the example of the world’s leading soccer clubs [22] and received support from an international training team (trainers, sports medicine physicians, physiotherapists, physiologists, and a nutritionist). The examined players had fulfilled rigorous criteria in the academy’s preliminary selection process, and they participated in a training program that was approved by the Hungarian Football Federation in consultation with UEFA and FIFA experts (including nutritionists). This is standard practice, and professional soccer clubs invest significant amounts of money to nurture elite players [37]. The players’ meals were carefully planned by nutritionists and were served at the academy’s cafeteria. In some cases, meals were planned individually (such as vegetarian meals). Therefore, excessive body mass, including excessive fat mass, in some young Hungarian soccer players could have resulted from additional food intake in-between meals. It is also possible that the selection criteria were not fully observed during the recruitment process due to a small number of candidates.

The absence of significant differences in muscle mass between the age groups also gives cause for concern. Muscle mass should be highest in the oldest group (17–18-year-olds), but the highest values of this parameter were noted in the 15-year-olds (difference of >1.4 kg relative to 14-year-olds; difference of >0.5 kg relative to 16-year-olds; difference of >0.8 kg relative to 17–18-year-olds). In males, muscle mass increases more rapidly past the age of 16 years [38], which is why the training programs at the ETO Football Academy have to be adapted to the needs of certain age groups to promote a proportional increase in the players’ muscle mass. The main emphasis should be placed on strength training and the players’ nutritional needs. Personal training options could also be considered. Resistance training should stimulate the development of endurance-strength abilities, not only an increase in muscle mass [39]. In dynamic sports that involve large muscles, such as soccer, it is generally assumed that VO_2max_ is primarily limited by maximal cardiac output [40]. From the practical point of view, special training interventions based on high-intensity interval training (HIIT) are recommended [5,41,42].

In general, an analysis of different playing positions (forwards, defenders, midfielders, and goalkeepers) demonstrated that anthropometric (biological age, body mass, body height, BMI, and PLX) and physiological parameters were higher in goalkeepers in the largest number of cases. In terms of anthropometric characteristics (body height, body mass, PLX, BMI and endomorphic somatotype), goalkeepers differed markedly from midfielders (six cases), defenders (four cases) and forwards (three cases).

### 4.2. Physiological Profile

The oldest soccer players scored significantly higher results in physical performance tests than the 14- and 15-year-olds, but significant differences should also be observed between 16-year-olds and 17–18-year-olds. The most talented 18-year-olds represent national teams or the best soccer clubs in the world [43]. Surprisingly, HR_rest_ values were lowest in 15-year-old Hungarian soccer players, which could be attributed to a more demanding training cycle than in other age groups. In well-trained athletes (in particular in endurance sports), HR_rest_ values can be as low as 50 bpm, i.e., typical of sinus bradycardia [44]. Meanwhile, only one player from the 17–18-year-old group achieved a lower value of HR_rest_ (47 bpm), and this parameter varied considerably within the age groups (differences between min and max values: 36 bpm in 14-year-olds, 18 bpm in 15-year-olds, 33 bpm in 16-year-olds, and 48 bpm in 17–18-year-olds). The lowest mean values of HR_max_ and the smallest differences between the minimum and maximum values of this parameter (a more homogeneous group) confirm the assumption that the 15-year-old players had been subjected to the most strenuous training regime. In turn, the highest mean values of HR_rest_ and the greatest differences between the minimum and maximum values of this parameter in the oldest players point to the greatest variations in motor abilities in 17–18-year-olds. These results suggest that the intensity and duration of the training load were probably insufficient in the oldest groups. It is possible that some players had quit the academy and were replaced by other athletes during the training program. The absence of significant differences in HR_max_ and ATP values across the age groups during motor tests could also imply that the entire training cycle lacked cohesion. In general, HR_max_ values should be highest in the youngest groups, although the players’ training experience should also be considered in this case. For this reason, training effects should be most evident in the oldest players who have the longest training experience. Players assigned different positions can also differ significantly in certain physiological characteristics in motor tests [45].

An analysis of rVO_2max_ and rVO_2_/AT values, which are regarded as the most important components of endurance performance [46], revealed certain training deficiencies (no training load progression due to age and training experience). Surprisingly, the mean values of rVO_2max_ and rVO_2_/AT were highest in the youngest group (57.6 and 51.2 mL/kg/min, respectively), and VO_2max_ values were significantly higher than in the 16-year-olds (46.8 mL/kg/min). Moreover, the 14-year-olds were also characterized by the highest maximal values of rVO_2max_ and rVO_2_/AT (68.3 and 60.8 mL/kg/min, respectively) and the greatest differences between minimal and maximal values (24.5 and 21.9 mL/kg/min, respectively), which points to considerable variations in aerobic thresholds in this age group. The mean VO_2max_ of elite soccer players generally ranges from 55 to 68 mL/kg/min and is influenced by the playing position [41]. However, values higher than 70 mL/kg/min have been also reported in modern soccer players [42,47].

Junior soccer players have lower VO_2max_ (<60 mL/kg/min) than seniors, with some exceptions. In a study by Helgerud et al. [42], VO_2max_ reached 64.3 mL/kg/min in juniors, whereas the mean value of VO_2max_ in the Hungarian national team (under 18) was determined at 73.9 mL/kg/min [48]. Therefore, the mean VO_2max_ values noted in 14-year-old Hungarian soccer players in 2021 (57.6 mL/kg/min) and the values reported more than 30 years earlier can be regarded as average, whereas the values reported in the two oldest groups were quite disappointing. The rank correlation analysis of the most successful teams in the Hungarian First Division Championship emphasizes the importance of VO_2max_ in soccer [48]. The presence of a strong correlation between VO_2max_ and distance covered during the match indicates that training regimes that raise VO_2max_ values should be adopted [49]. According to Smaros [50], VO_2max_ was strongly correlated with the total distance covered in the game (r = 0.89), but it also influenced the number of sprints attempted during a match. Recent research has shown that an 11% improvement in the mean VO_2max_ of youth soccer players over a period of 8 weeks resulted in a 20% increase in the total distance covered during competitive matches, a 23% increase in ball possession, and a 100% increase in the number of sprints performed by each player [42]. Some studies reported that young soccer players had similar VO_2max_, but lower running economy than adults when expressed in mL/kg/min [51,52].

When players differ considerably in physical performance parameters, trainers are less able to plan effective training sessions and select the best players for the game. Although untrained subjects are limited peripherally, trained individuals are primarily limited centrally, and maximal stroke volume is regarded as the major limiting factor for VO_2max_ [40,53]. Zhou et al. [54] reported increased stroke volume to the level of VO_2max_. These findings were not confirmed in the present study, where HR_max_ values did not differ significantly (*p* > 0.05) between the compared age groups. This observation provides yet another argument that the training program at the ETO Soccer Academy requires a critical assessment. Match running performance based on fixed speed thresholds increases concomitantly with age [13,29]. In theory, when conditioning sessions are designed based on the actual match load, running capability should be adjusted progressively to age across all categories [15,55]. In the present study, the above observation was confirmed by the results of the Yo-Yo test and the 20-m sprint test, but no significant differences (*p* > 0.05) were noted in sprint performance over short distances (5 m and 10 m). Some authors have argued that younger players have lower running capabilities due to lower technical-tactical skills [13,15,56], which is why generalizations regarding changes in match running performance should be avoided [29].

### 4.3. Analysis of Player Profiles Based on Position

A review of the literature on young soccer players indicates that match running performance is dependent on position [57]. Running performance assessments based on time–motion analyses (TMA) are presently considered a fundamental part of youth development [57]. However, the current study revealed discrepancies in the training program of young Hungarian soccer players; therefore, the question that remains to be answered is whether the differences presented in the literature [14,15,58,59] are meaningful and can be used to inform subsequent position-specific training regimens [60].

In terms of physiological characteristics, goalkeepers scored significantly higher than forwards (10 cases), midfielders (nine cases), and defenders (three cases). These differences were observed in tests assessing lower limb strength (NB max dominant and non-dominant leg, NB avg non-dominant leg, FF max dominant and non-dominant adductors, and FF max dominant and non-dominant abductors), sprint performance over 5 m and 10 m, and agility tests (Illinois test with and without a ball). Agility is regarded as one of the most important determinants of soccer performance; therefore, goalkeepers and other Hungarian soccer players should also develop their motor skills during training [61]. Soccer players with a high agility profile are more likely to perform better during repeated high-speed activities and to make faster decisions during critical moments of a match [62,63]. The above findings were confirmed by a study comparing the performance of soccer players younger than 15 in 10-m sprint and vertical jump tests, where elite and sub-elite players clearly outperformed the remaining contestants. However, it should be noted that the players’ ability to change direction during the match depends not only on physical capacity, but also on perceptual and decision-making factors (reaction skills) that are largely genetically conditioned [64]. Goalkeepers received significantly lower scores than other players (forwards, defenders and midfielders, *p* < 0.001) only in the Yo-Yo test. Goalkeepers play a special role in modern soccer because they are not only required to defend the goal, but also to actively cooperate with their partners in defending and attacking situations [65]. Therefore, their motor activity changes in various tactical situations [66]. The activity profile of goalkeepers was examined in a limited number of studies [67,68]. Di Salvo et al. [69] analyzed the distance covered by goalkeepers from English Premier League teams, whereas Condello et al. [70] conducted a similar study on non-professional goalkeepers. Research on Polish goalkeepers demonstrated they were characterized by lower levels of motor activity than other players and covered an average distance of around 4730 ± 835 m during a game, including 3441 ± 597 m (73%) walking, 1175 ± 371 m (25%) running, and 114 ± 55 m sprinting (2%) [65]. The distance covered by the fastest goalkeepers was several times shorter than that covered in other playing positions [66,71,72]. These observations could explain why Hungarian goalkeepers scored significantly lower results in the Yo-Yo test than other players. However, it remains unknown whether Hungarian goalkeepers should significantly outperform other players in terms of lower limb strength. The absence of significant differences in the values of HR_rest_, HR_max_, and ATP is also surprising because goalkeepers generally cover shorter distances than other players. Therefore, the absence of differences in the values of rVO_2max_ and rVO_2_/AT should be also interpreted with caution because these parameters should be significantly lower in goalkeepers than in field players. Strøyer et al. [73] observed higher VO_2max_ values in midfielders and forwards than in defenders (65 vs. 58 mL/kg/min, respectively, for young elite soccer players at the end of puberty, i.e., 14 years of age). However, no such differences were noted in the present study. Relative and biological age was significantly higher in goalkeepers, and these parameters should probably also be considered in comparative analyses. Leyhr et al. [43] reported a trend toward a better holistic rating of early born players for current and future performance levels. Other authors found that relative age increased at higher performance levels in youth soccer [74,75].

### 4.4. Strengths and Limitations

The main strength of this study was the creation of preliminary anthropometric and physiological profiles of Hungarian soccer players. These profiles can facilitate the selection process, and they can be used to identify and eliminate potential problems in the training cycle. Anthropometric and physiological profiles constitute robust indicators for selecting the best players who, upon completing the full training cycle in the academy, should join the top Hungarian soccer clubs and, ideally, the national soccer team. A comparison of anthropometric and physiological parameters in differently aged players, in players with various training experience, and in players assigned different positions, revealed certain problem areas in the training program of the ETO Soccer Academy.

The main limitation of this study was a relatively low number of players classified into age groups and position groups (this applies mainly to goalkeepers). The development of comprehensive profiles, such as body type profiles, requires a large number of standardized variables and a sufficiently large sample. Unfortunately, other Hungarian soccer academies were reluctant to share such data, which could be explained by their autonomous status or the fear that the relevant information would shed light on the real reasons behind the lack of progress in soccer development in Hungary. The variables were transformed into Z-scores to compensate for these limitations and to compare differently aged athletes playing in the same positions. Another limitation is the fact that the study was conducted during the pre-season preparatory period when players often present inferior body composition and decreased performance levels. From this point of view, repeated measurements at different times of the season would provide valuable material for detailed comparative analyses.

## 5. Conclusions

The findings of this study indicate that basic somatic parameters in the examined soccer players increased with age, which is consistent with natural biological development processes. Athletes with an ectomorphic body type were predominant in all age groups of Hungarian soccer players. The average values of anthropometric, body composition, physiological and performance variables were within normal limits in all age groups, which suggests that the selection process was appropriate. Surprisingly, across the age groups, the mean values of rVO_2max_ and rVO_2/AT_ were highest in the youngest, 14-year-old players (57.6 and 51.2 mL/kg/min, respectively; smallest range of values). This study also confirmed significant differences in the anthropometric and physiological characteristics of various playing positions; the highest values of these parameters were observed in goalkeepers. Goalkeepers scored higher in lower limb strength (NB max dominant and non-dominant leg, NB avg non-dominant leg, FF max dominant and non-dominant adductors, and FF max right and left abductors), sprint performance over 5 and 10 m, and agility tests (Illinois test with and without a ball). Goalkeepers scored significantly lower results than other players (forwards, defenders, and midfielders) only in the Yo-Yo test. Considering the practical applications, the current findings suggest that national classification standards based on the anthropometric and physiological characteristics of players attending all eleven Hungarian soccer academies should be developed. Such standards would support the rapid analyses and comparisons of soccer players within age groups based on selected strength and endurance parameters as well as the general profiles of soccer players (GPSP). Finally, in-depth insights into the anthropometric and physiological parameters of soccer players relative to the age group, competitive level and playing position would help practitioners to (a) provide individualized practice, in an attempt to evaluate and develop soccer-specific skills relative to these factors, and (b) optimize player performance.

## Figures and Tables

**Table 1 ijerph-19-11041-t001:** (**a**) Descriptive and comparative analysis of chronological and biological age and the anthropometric characteristics of soccer players in the four analyzed age categories. (**b**) Descriptive and comparative analysis of the physiological and performance parameters of soccer players in the four analyzed age categories.

Parameter	Age Category [years]												Difference		Effect Size
	14 (*n* = 20)			15 (*n* = 16)			16 (*n* = 22)			17–18 (*n* = 23)					
	Mean	SD	Min–Max	Mean	SD	Min–Max	Mean	SD	Min–Max	Mean	SD	Min–Max	*F*	*p*	ω^2^
(**a**)															
Chronological age [years]	14.46 ^2.3.4^	0.34	14.01–14.99	15.63 ^1.3.4^	0.27	15.13–15.99	16.64 ^1.2.4^	0.20	16.13–16.99	17.97 ^1.2.3^	0.47	17.09–18.92	407.82	<0.001	0.94
Biological age [years]	14.40 ^2.3.4^	0.57	12.86–15.49	15.27 ^1.3.4^	0.62	14.35–16.31	16.58 ^1.2.4^	0.44	15.68–17.31	17.92 ^1.2.3^	0.46	16.75–18.34	184.80	<0.001	0.87
Body height [cm]	171.83 ^4^	9.58	154.6–188.2	176.66	7.05	166.4–188.1	177.72	9.23	160.5–194.3	182.38 ^1^	5.14	170.7–192.2	6.35	<0.001	0.17
Body mass [kg]	58.47 ^3.4^	8.92	38.15–80.25	61.52 ^4^	7.60	51.10–81.52	66.97 ^1^	10.38	50.45–91.80	72.16 ^1.2^	6.90	61.55–91.40	10.41	<0.001	0.26
BMI [kg/m^2^]	19.72 ^4^	1.85	15.96–23.68	19.69 ^4^	1.90	17.40–24.09	21.10	1.85	17.47–24.59	21.70 ^1.2^	1.93	18.97–27.32	5.81	<0.001	0.15
Fat mass [%]	8.41 ^4^	3.18	4.5–14.8	7.61 ^3.4^	1.52	4.4–10.0	10.37 ^2^	3.07	5.6–17.1	11.67 ^1.2^	4.01	5.9–23.9	6.66	<0.001	0.22
Muscle mass [%]	42.12	2.23	37.2–44.6	43.56	1.89	41.3–49.0	43.05	1.48	40.2–45.9	42.76	2.02	38.2–47.0	1.79	0.156	0.03
Endomorphic	2.45	0.79	1.42–4.37	2.27	0.52	1.57–3.06	2.54	0.80	1.13–4.45	2.75	0.91	1.42–5.37	1.26	0.294	0.01
Mesomorphic	3.01	1.18	0.98–5.26	2.51	1.22	1.29–5.64	3.20	1.01	0.52–5.55	3.14	0.96	1.51–6.27	1.48	0.225	0.02
Ectomorphic	3.91	1.07	1.54–5.49	4.33	1.15	1.54–5.82	3.53	1.04	1.18–6.04	3.50	1.04	0.50–5.22	2.42	0.072	0.05
PLX	80.01 ^3.4^	4.89	68.2–87.0	82.78 ^4^	3.51	78.1–90.6	85.58 ^1^	4.71	79.2–97.3	87.69 ^1.2^	3.08	83.2–93.3	13.60	<0.001	0.32
NB max D leg [N/m]	301.35 ^3.4^	63.45	170.5–475.3	318.56 ^3.4^	46.57	257.0–417.8	376.76 ^1.2^	39.13	284.8–449.5	396.70 ^1.2^	37.32	331.0–475.0	19.21	<0.001	0.40
NB max ND leg [N/m]	311.53 ^3.4^	66.96	158.8–479.0	328.67 ^3.4^	45.37	248.8–423.0	390.61 ^1.2^	50.15	299.8–479.8	401.87 ^1.2^	38.41	329.0–503.1	15.72	<0.001	0.35
NB-IBM [%]	−2.41	7.34	−20.4–9.8	−3.03	5.20	−15.2–4.8	−3.05	8.65	−14.5–15.5	−1.22	6.31	−19.1–10.4	0.31	0.816	−0.02
NB avg D leg [N/m]	290.13 ^3.4^	69.3	170.5–460.3	319.64 ^3.4^	48.34	240.2–423.0	368.74 ^1.2^	37.92	284.3–433.6	370.43 ^1.2^	43.87	290.1–446.0	12.37	<0.001	0.30
NB avg ND leg [N/m]	301.93 ^3.4^	75.29	158.8–476.6	309.64 ^3.4^	51.27	235.7–417.8	380.40 ^1.2^	49.30	283.8–486.4	375.13 ^1.2^	45.12	291.0–447.0	11.15	<0.001	0.27
(**b**)															
FF max D AD [N/m]	299.23 ^2.3.4^	68.88	166.1–444.0	360.44 ^1.3^	60.85	264.8–458.3	422.89 ^1.2^	64.01	272.0–526.0	413.43 ^1^	58.63	333.1–512.0	16.88	<0.001	0.37
FF max ND AD [N/m]	289.02 ^2.3.4^	66.39	136.0–416.0	346.25 ^1.3.4^	54.43	236.8–433.7	412.34 ^1.2^	56.93	296.0–515.5	402.26 ^1.2^	57.09	323.0–509.2	19.37	<0.001	0.40
FF add. limb [%]	5.84	5.23	0.3–18.1	3.81	5.68	−10.1–15.0	3.60	4.47	−3.7–14.6	3.48	3.20	0.0–12.2	1.18	0.323	0.01
FF max D ABD [N/m]	287.25 ^3.4^	51.40	184.0–369.3	319.38 ^3.4^	51.19	253.4–437.2	382.85 ^1.2^	51.72	268.5–532.7	404.17 ^1.2^	44.80	327.0–485.1	24.93	<0.001	0.47
FF max ND ABD [N/m]	283.55 ^3.4^	43.20	200.0–387.0	318.67 ^3.4^	49.58	245.3–451.0	372.93 ^1.2^	48.23	301.0–496.2	392.91 ^1.2^	42.01	332.0–478.3	25.11	<0.001	0.47
FF ABD limb [%]	4.39 ^2^	3.24	0.2–10.0	0.10 ^1^	5.86	−10.1–12.6	2.95	4.41	-4.8–11.6	3.78	3.73	0.0–13.9	3.35	0.023	0.08
5 m speed [s]	1.115	0.118	0.94–1.32	1.105	0.107	0.92–1.30	1.113	0.114	0.83–1.25	1.035	0.125	0.86–1.30	2.36	0.078	0.05
10 m speed [s]	1.892	0.128	1.63–2.12	1.879	0.115	1.68–2.05	1.838	0.102	1.61–1.98	1.807	0.137	1.57–2.07	2.14	0.102	0.04
20 m speed [s]	3.205 ^4^	0.170	2.68–3.52	3.206 ^4^	0.144	2.94–3.41	3.113	0.147	2.82–3.50	3.067 ^1.2^	0.156	2.80–3.42	4.03	0.010	0.10
505 D leg [s]	2.656 ^2.3.4^	0.208	2.34–2.96	2.421 ^1^	0.098	2.21–2.57	2.364 ^1^	0.108	2.21–2.59	2.478 ^1.3^	0.107	2.27–2.71	16.97	<0.001	0.37
505 ND leg [s]	2.701 ^2.3.4^	0.251	2.45–3.50	2.504 ^1^	0.249	2.26–2.70	2.480 ^1^	0.102	2.26–2.70	2.470 ^1^	0.161	2.15–2.81	6.41	<0.001	0.17
IAGT with ball [s]	20.249 ^4^	0.983	18.21–22.01	19.736	1.202	18.65–23.79	19.765 ^4^	1.648	17.05–24.12	18.706 ^1.3^	0.977	17.02–20.86	6.01	0.001	0.17
IAGT without ball [s]	15.772 ^3.4^	0.523	14.98–17.46	15.484 ^4^	0.504	14.77–16.89	15.090 ^1.4^	0.416	14.21–16.02	14.495 ^1.2.3^	0.848	13.14–16.05	17.55	<0.001	0.38
Yo-Yo test [m]	1889.0 ^4^	393.4	1040–2520	1932.5 ^4^	269.2	1480–2400	2187.3	459.0	1120–3160	2495.7 ^1.2^	653.4	840–3520	7.04	<0.001	0.18
HR_rest_ [bpm]	72.65 ^2^	8.99	60–96	63.50 ^1.3.4^	4.99	54–72	76.27 ^2^	8.47	56–89	77.52 ^2^	12.23	47–95	8.22	<0.001	0.21
HR_max_ [bpm]	196.65	7.23	185–213	194.19	7.08	178–205	196.23	7.69	179–210	195.57	7.21	182–210	0.38	0.768	−0.02
rVO_2max_ [mL/kg/min]	57.62 ^3^	8.12	43.8–68.3	56.34	4.76	47.6–64.6	52.56 ^1^	4.64	45.1–62.7	56.64	3.11	49.9–65.0	3.56	0.018	0.09
rVO_2_/AT [mL/kg/min]	51.20 ^3^	7.24	38.9–60.8	50.14	4.23	42.4–57.5	46.75 ^1^	4.11	40.15–55.8	50.31	2.98	44.4–57.9	3.47	0.020	0.08
ATP [bpm]	183.18	6.69	172–198	180.59	6.59	166–191	182.57	7.16	166–195	182.25	6.98	169–195	0.46	0.721	−0.02

***Notes:*** BMI—body mass index, PLX—plastic index, NB max D leg—Nord Board maximal dominant leg, NB-IMB—Nord Board Imbalance, NB avg D leg—Nord Board average dominant leg, NB avg ND leg—Nord Board average non-dominant leg, FF max D AD—Force Frame maximal dominant leg, FF max ND AD—Force Frame maximal non-dominant leg, FF add. Limb—Force Frame adductor limb, FF max D ABD—Force Frame dominant abductor, FF max ND ABD—Force Frame maximal non-dominant abductor, FF ABD limb—Force Frame abductor limb, 5, 10, 20 m speed—Sprint Performance linear sprinting, 505 D and ND leg—Change of direction test, IAGT with and without ball—Illinois change of direction (*COD*), Yo-Yo-test-Intermittent Recovery Test Level 1, HR_rest_—resting pulse, HR_max_—maximal pulse, rVO_2max_—relative aerobe capacity, rVO_2_/AT—anaerobic threshold, ATP—anaerobic threshold pulse. ^1^—14-year-old, ^2^—15-year-old, ^3^—16-year-old, ^4^—17–18-year-old.

**Table 2 ijerph-19-11041-t002:** (**a**) Descriptive and comparative analysis of the Z-scores of chronological and biological age and the anthropometric characteristics of soccer players assigned to different playing positions. (**b**) Descriptive and comparative analysis of the Z-scores of the physiological and performance parameters of soccer players assigned to different playing positions.

Parameter	Playing Position								Difference		Effect Size
	1 (*n* = 23)		2 (*n* = 29)		3 (*n* = 22)		4 (*n* = 7)				
	Mean	SD	Mean	SD	Mean	SD	Mean	SD	*F*	*p*	ω^2^
(**a**)											
Biological age [years]	0.03	1.12	0.09	0.76	−0.44 ^4^	1.06	0.87 ^3^	0.17	3.72	0.015	0.07
Body height [cm]	−0.11 ^4^	1.00	0.01 ^4^	0.85	−0.35 ^4^	0.89	1.42 ^1.2.3^	0.40	7.27	<0.001	0.16
Body mass [kg]	−0.08 ^4^	0.95	0.14 ^3.4^	0.83	−0.58 ^2.4^	0.70	1.51 ^1.2.3^	0.80	11.69	<0.001	0.24
BMI [kg/m^2^]	−0.01	1.02	0.16	1.07	−0.47 ^4^	0.55	0.82 ^3^	0.98	3.95	0.011	0.10
Fat mass [%]	0.01	0.76	0.00	1.12	−0.24	0.77	0.71	1.40	1.71	0.111	0.04
Muscle mass [%]	−0.09	0.92	0.06	1.03	0.01	0.97	0.02	1.22	0.10	0.904	−0.03
Endomorphic	0.03	0.65	−0.04 ^4^	1.13	−0.11 ^4^	0.84	1.04 ^2.3^	1.27	2.84	0.043	0.06
Mesomorphic	−0.03	0.89	0.24	0.91	−0.24	0.73	−0.14	0.41	1.48	0.530	−0.01
Ectomorphic	−0.02	1.04	−0.15	1.14	0.27	0.71	−0.11	0.81	0.81	0.441	−0.00
PLX	−0.15 ^4^	0.94	0.09 ^4^	0.81	−0.48 ^4^	0.74	1.66 ^1.2.3^	0.67	12.34	<0.001	0.23
NB max D leg [N/m]	−0.13	0.83	0.17	1.02	−0.38 ^4^	0.83	0.92 ^3^	1.15	4.01	0.010	0.09
NB max ND leg [N/m]	0.01	1.05	0.09	0.92	−0.38 ^4^	0.77	0.80 ^3^	1.22	2.91	0.040	0.07
NB-IBM [%]	−0.08	0.96	0.12	1.12	−0.11	0.94	0.13	0.54	0.31	0.835	−0.03
NB avg D leg [N/m]	−0.20 ^4^	0.67	0.10	1.05	−0.20 ^4^	0.99	0.88 ^1.3^	1.19	2.76	0.048	0.07
NB avg ND leg [N/m]	−0.15	0.85	0.16	0.95	−0.26	0.95	0.65	1.37	2.09	0.052	0.06
(**b**)											
FF max D ADD [N/m]	−0.26 ^4^	1.05	0.13	1.06	−0.20 ^4^	0.69	0.96 ^1.3^	0.56	3.57	0.018	0.09
FF max ND ADD [N/m]	−0.28 ^4^	1.05	0.13	0.99	−0.15	0.76	0.85 ^1^	0.93	2.94	0.038	0.08
FF ADD limb [%]	0.15	1.05	0.03	1.09	−0.30	0.59	0.28	1.17	1.08	0.370	0.00
FF max D ABD [N/m]	0.06 ^4^	1.09	0.10 ^4^	0.78	−0.56 ^4^	0.83	1.16 ^1.2.3^	0.62	7.06	<0.001	0.11
FF max ND ABD [N/m]	−0.01 ^4^	1.01	0.08 ^4^	0.84	−0.53 ^4^	0.77	1.36 ^1.2.3^	0.70	8.54	<0.001	0.14
FF ABD limb [%]	0.23	1.02	0.03	1.01	−0.11	0.99	−0.52	0.53	1.19	0.288	0.01
5 m speed [s]	−0.27 ^4^	1.12	0.16	0.93	−0.18	0.80	0.82 ^1^	0.85	2.92	0.040	0.05
10 m speed [s]	−0.25 ^4^	1.12	0.02	0.83	−0.09 ^4^	0.93	1.03 ^1.3^	0.70	3.44	0.021	0.05
20 m speed [s]	−0.22 ^4^	1.06	−0.02	0.91	−0.04	0.93	0.91 ^1^	0.78	2.56	0.051	0.02
505 Right leg [s]	−0.17	1.11	0.08	0.92	0.04	1.02	0.10	0.74	0.31	0.955	−0.03
505 Left leg [s]	−0.17	0.88	0.08	0.75	−0.16	1.19	0.77	1.21	1.99	0.788	−0.02
Illinois with ball [s]	−0.30 ^4^	0.95	−0.06 ^4^	0.65	−0.21 ^4^	0.65	1.89 ^1.2.3^	1.18	14.86	<0.001	0.31
Illinois without ball [s]	−0.35 ^4^	0.88	0.14	0.78	−0.09 ^4^	1.08	0.97 ^1.3^	1.19	3.81	0.013	0.07
Yo-Yo test [m]	−0.02 ^4^	0.84	0.23 ^4^	0.76	0.11 ^4^	1.67	−1.22 ^1.2.3^	0.85	4.85	0.004	0.08
HR_rest_ [bpm]	0.19	1.00	0.05	0.96	0.00	0.79	−0.85	1.28	2.17	0.207	0.02
HR_max_ [bpm]	0.01	0.81	−0.03	1.08	0.20	1.05	−0.52	0.84	0.96	0.412	−0.00
rVO_2max_ [mL/kg/min]	0.21	1.07	0.01	1.06	−0.22	0.87	−0.03	0.93	0.68	0.528	−0.01
rVO_2_/AT [mL/kg/min]	0.21	1.05	0.00	1.04	−0.21	0.86	−0.04	0.90	0.68	0.507	−0.01
ATP [bpm]	0.01	0.79	0.02	1.10	0.25	1.04	−0.44	0.82	0.91	0.418	−0.00

***Notes:***^1^—forward, ^2^—defender, ^3^—midfielder, ^4^—goalkeeper.

## Data Availability

The data used to support the findings of this study are restricted by the Ethics Committee of the University of Warmia and Mazury in Olsztyn (UWM), Poland in order to protect the participants’ privacy. The data are available from R.P., E-mail: podstawskirobert@gmail.com for researchers who meet the criteria for access to confidential data.

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
