# Peer review of "Anthropometric and Physiological Profiles of Hungarian Youth Male Soccer Players of Varying Ages and Playing Positions: A Multidimensional Assessment with a Critical Approach"

_ijerph, 2022, doi:10.3390/ijerph191711041_

Round 1

Reviewer 1 Report

This article aims to identify anthropometric and physiological profiles of male soccer players according to age and playing positions.

The article addresses an interesting topic, however there is a limitation that the data were collected in a single football club, so it cannot be extrapolated to Hungary.

On the other hand, the sample is composed of semi-amateur players, of a competitive level that is not elite. This fact may affect the interest of the results obtained. It's more of a case study, for the club itself to reflect on.

However, sharing results with the scientific community is welcome, as it may be important for researchers working with other similar clubs.

In the introduction, it was important to report that it refers to the literature on:

1. the evolution of anthropometric characteristics with age in young footballers;

2. the evolution of physiological competences with age in young footballers;

3. the relationship between anthropometric characteristics and playing position;

4. the relationship between physiological skills and playing position;

This clarification would help clarify the research hypotheses of the study.

The methodology used is well described and the results are clearly presented.

The discussion would be more interesting if there were more comparisons with normative values, or comparable values existing in the literature (the authors already make this comparison but could go further).

Authors should include study limitations and emphasize the practical applicability of this investigation

Author Response

Dear Reviewer,

The response is on file.

Best regards,

Robert Podstawski

Reviewer 2 Report

Dear Authors

You have written interesting research. However several parts need to be addressed for greater clarity and reproducibility.

The abstract conclusion is poorly written and does not present anything new. Please rewrite.

The introduction is poorly written. The majority of the introduction is focused on senior players and almost nothing on youth soccer players. There are several studies out there from other countries that have researched this particular problem. Therefore, please update the introduction with the main focus on youth soccer players.

Participants - How was your sample size determined (G*Power or any other method)? Report

Line 102 - Define elite soccer players 

What were their activity levels - how many trainings per week, duration of trainings? report

Report the number of participants per position in the age groups.

Line 132 - was the researcher performing the measurements ISAK accredited? report

Report all variables were calculated via manual anthropometry  (in table 1 you also report somatotype)

Line 136 - remove )

Report which body composition variables were taken forward with Inbody 720. Additionally, please explain what instructions were given prior to bioimpedance measurements as this can largely affect mesurement results. I suggest looking at the instructions published for Inbody 720 ( Experimental Procedure section) : https://doi.org/10.3390/biology10111199

Please report which anthropometric instruments have been used in measurements.

Please report what was the exact order of all tests and what was the break in between the tests.

Line 165 - please explain this distinct peak

VALD testing - Please report how many repetitions did they perform, what was the break between them and what result and which variables were taken to further analysis

Line 183 - add reference for the equation

You discuss HR at rest however you never report where or how was it measured. Elaborate

Results - Table 1 - All abbreviations need to be explained. Amend accordingly

Limitations section should be extended: players in the preparation period, a small number of players per position, especially goalkeepers.

Line 566 - Why do you report max values of 23.9% of body fat and 27.3 BMI in the oldest group if the mean values are much lower? This is perhaps your opinion that there are some athletes that need to lower those values, however, this needs to be looked in consideration to muscle mass % which is again at good levels. Therefore, make the conclusion on mean values and not on max values. Please rewrite your conclusions.

I don't see any practical application of your study in the conclusion and how can these results be used in practice by coaches? Please amend

Overall an interesting paper; however, it still needs a substantial upgrade from the authors. Therefore I recommend a major revision.

Kind regards

Author Response

(The authors gave the same response as above.)

Round 2

Reviewer 2 Report

Dear Authors

Thank you for addressing some of my comments.

However, some of them have not been adequately addressed and need some more work.

The introduction has not been updated as requested - Therefore, update the introduction with the main focus on youth soccer players!

You have added references 28 and 29 which have no connection to youth players as their sample is comprised of adult players  - so what is the point? Please add youth players' studies that evaluated similar variables as you and comment/compare them in the discussion.

The main rationale is not backed up by your introduction as I mentioned it is all about adult players and almost nothing focused on youth players.

Please then report post hoc calculation with G*Power software of your sample power for your test as your description of sample acquisition is not sufficient for this level of journals.

Line 109 - trainee-level elite  / these two words don't go together how can someone that is a trainee be elite?

Your explanation about elite-level youth athletes is not convincing. It raises even more questions. So you had an extra criterion regarding the quality, low and high-level players? Elaborate and add this if necessary in the paper

Add info in the paper that the anthropometrist was Isak accredited and add the level of accreditation.

You not finding the reference for one of the most used bilateral asymmetries equations is not what I would expect from such a group of researchers. Nonetheless, you could look at the following reference : DOI: 10.1519/SSC.0000000000000264

https://journals.lww.com/nsca-scj/Fulltext/2016/12000/Asymmetries_of_the_Lower_Limb__The_Calculation.3.aspx

HR-at rest - please add the model of the HR monitor that was used.

The conclusion was not amended as stated by the authors as the numbers are still the same!!!

Why do you report max values of 23.9% of body fat and 27.3 BMI in the oldest group if the mean values are much
lower? This is perhaps your opinion that there are some athletes that need to lower those values, however, this needs to be looked in consideration to muscle mass %
which is again at good levels. Therefore, make the conclusion on mean values and not on max values. Please rewrite your conclusions.

Also, the conclusion needs to be shortened and only the so-called ''study highlights'' need to be present.

The quality of paper has improved, however, there is still more room for improvement. 

Therefore I again recommend major revision as the paper has good potential.

Kind regards

Author Response

Dear Reviewer,

Attached you will find our response to the review.

Best regards,

Robert Podstawski
